# Multi-Agent Retrieval-Augmented Framework for Evidence-Based Counterspeech Against Health Misinformation

**Anirban Saha Anik** *
University of North Texas
AnirbanSahaAnik@my.unt.edu

**Xiaoying Song** *
University of North Texas
XiaoyingSong@my.unt.edu

**Elliott Wang**
University of North Texas
ElliottWang@my.unt.edu

**Bryan Wang**
University of North Texas
BryanWang2@my.unt.edu

**Bengisu Yarimbas**
University of North Texas
bengisuyarimbas@gmail.com

**Lingzi Hong**
University of North Texas
Lingzi.Hong@unt.edu

## Abstract

Large language models (LLMs) incorporated with Retrieval-Augmented Generation (RAG) have demonstrated powerful capabilities in generating counterspeech against misinformation. However, current studies rely on limited evidence and offer less control over final outputs. To address these challenges, we propose a Multi-agent Retrieval-Augmented Framework to generate counterspeech against health misinformation, incorporating multiple LLMs to optimize knowledge retrieval, evidence enhancement, and response refinement. Our approach integrates both static and dynamic evidence, ensuring that the generated counterspeech is relevant, well-grounded, and up-to-date. Our method outperforms baseline approaches in politeness, relevance, informativeness, and factual accuracy, demonstrating its effectiveness in generating high-quality counterspeech. To further validate our approach, we conduct ablation studies to verify the necessity of each component in our framework. Furthermore, cross evaluations show that our system generalizes well across diverse health misinformation topics and datasets. And human evaluations reveal that refinement significantly enhances counterspeech quality and obtains human preference.

## 1 Introduction

The rapid spread of misinformation on social media has become a significant concern, particularly in health-related topics. Health misinformation about disease prevention and untested treatments spreads widely, confusing the public and undermining trust in medical institutions (Chou et al., 2021; Cinelli et al., 2020). Consequently, individuals hesitate to follow public health guidelines, and some even resort to harmful self-medication practices (Pennycook et al., 2020), underscoring the emergence of mitigating health misinformation. Recent studies prove that social media users can help with misinformation debunking (Chuai et al., 2024; Caulfield et al., 2020), who can rapidly respond, share factual information and correct misinformation, extending the impact of debunking efforts beyond official channels. However, they often struggle to access reliable evidence and are limited by the large volume and complexity of health misinformation (Mourali & Drake, 2022). Therefore, empowering users to effectively counter misinformation is crucial (He et al., 2023; Swire-Thompson et al., 2020).

Large language models (LLMs) are powerful in response generation and are widely used in mitigating misinformation (Proma et al., 2025; Chen & Shu, 2024; Saha et al., 2024; Hong et al., 2024). However, LLMs are not always reliable and can produce hallucinations, prompt-

---

*These authors contributed equally to this work and are considered joint first authors.

ing researchers to integrate external knowledge to counter misinformation (Bondarenko & Viehweger, 2024). Retrieval-Augmented Generation (RAG) has been employed to incorporate external knowledge, facilitating the generation of evidence-based counterspeech (Ni et al., 2025; Yue et al., 2024; Liu et al., 2024). Various studies typically rely on **static knowledge** (Chung et al., 2021; Guu et al., 2020), which refers to verified knowledge that is relatively stable and does not frequently change (Shrinivasan & Razniewski, 2022). For example, the U.S. Surgeon General's Advisory on Building a Healthy Information Environment [1] presents established public health principles and guidelines for addressing health misinformation. Recently, **dynamic knowledge** has been increasingly incorporated into response generation (Yeginbergen et al., 2025; Komeili et al., 2022; Nakano et al., 2021). This kind of knowledge usually refers to information that is continually updated and responsive to evolving situations. For instance, *"The U.S. Department of Health and Human Services (HHS) has announced that it will produce 4.8 million doses of the H5N1 avian flu vaccine to prepare for a possible pandemic."* This development has been reported by reliable sources such as CIDRAP[2], CBS News[3], and AJMC[4]. Although recent studies have incorporated various forms of knowledge, most existing methods utilize only one type of knowledge: either static or dynamic, ignoring the potential benefits of interacting with both.

Incorporating both static and dynamic knowledge is essential in mitigating health misinformation. Firstly, they are complementary to each other. Although the static knowledge is reliable, it has the limitation that it can not quickly adapt to emerging health misinformation or recent updates. In contrast, dynamic evidence can adapt rapidly to new contexts, but may lack nuance or rigorous validation, potentially weakening the effectiveness of counterspeech (Braun et al., 2024). Therefore, integrating both forms of knowledge allows static knowledge, such as established guidelines, to help filter out low-quality information, while dynamic knowledge ensures the latest development (Nezafat & Samet, 2024). Secondly, incorporating both types of evidence helps build public trust. Individuals are more likely to follow advice relying on longstanding, peer-reviewed research while simultaneously staying alert to contemporary changes (Yeginbergen et al., 2025; Drolsbach et al., 2024).

In our study, we design a multi-source knowledge framework that encompasses both static and dynamic evidence. The static knowledge base provides reliable, focused information, built through careful manual curation. Meanwhile, the dynamic knowledge base continuously integrates real-time updates using a web search API, ensuring that the framework remains responsive to evolving health information. However, incorporating large volumes of evidence may lead to information overload, resulting in responses that are less accessible to users. To address this, we design a refined module to post-edit responses, ensuring they are clear, concise, and user-friendly (Madaan et al., 2023; Deng et al., 2023). In our study, we propose a **Multi-Agent Retrieval-Augmented Framework** that integrates various components to achieve effective counterspeech generation. Multiple agents are designed to perform various tasks, including retrieving evidence from static and dynamic knowledge bases, summarizing evidence, generating responses, and refining them for clarity and accessibility.

Our contributions in this study include:

- **Multi-Agent Retrieval-Augmented Framework Specialized for Counterspeech Generation Against Health Misinformation**
  This pipeline presents a modular multi-agent framework tailored for generating effective counterspeech in response to health misinformation. It integrates specialized sub-agents responsible for: (1) static and dynamic retrieval, (2) evidence summarization, (3) counterspeech generation, and (4) iterative response refinement. Each agent functions autonomously yet collaboratively within a scalable architecture, enhancing the quality of the generated counterspeech.

---

[1] https://www.ncbi.nlm.nih.gov/
[2] https://www.cidrap.umn.edu/
[3] https://www.cbsnews.com/
[4] https://www.ajmc.com/

- **Integration of Static and Dynamic Knowledge**
  The framework combines both static (e.g., curated medical guidelines) and dynamic (e.g., real-time web search) evidence to generate counterspeech, enabling responses that are both authoritative and up-to-date. To our knowledge, this combination has not been explored previously, especially in the context of generating counterspeech to health misinformation. Our experiments demonstrate the importance of incorporating both static and dynamic knowledge.

- **Ablation Studies with Prompting Strategy Comparison**
  Comprehensive ablation studies evaluate the contribution of each agent in the pipeline. The system is tested under different prompting strategies (e.g., Guided Prompting vs. Chain-of-Thought), demonstrating that each agent remains essential regardless of the prompting method used. Such testing has not been implemented or reported in previous research on counterspeech generation.

- **A Curated Dataset on Diverse Health Misinformation Topics**
  A new dataset is introduced, featuring health misinformation posts related to COVID-19, influenza, and HIV, sourced from Reddit and annotated through a combination of classifier-assisted annotation and expert labeling. **This dataset is valuable for health misinformation research and is publicly available on GitHub [5].**

## 2 Related Work

### 2.1 Evidence-based Misinformation Mitigation

Recently, several studies have leveraged LLMs to produce contextually relevant and effective responses by utilizing their own knowledge base (Mou et al., 2024; Lodovico Molina et al., 2024). However, due to the limitation of LLMs' knowledge base, they produced hallucinations and fabricated information (Huang et al., 2025; Rawte et al., 2023). Retrieval-augmented generation (RAG) was proposed to introduce external evidence for producing counterspeech (Ni et al., 2025; Yue et al., 2024; Leekha et al., 2024), improving the factual accuracy of the content generated. This method primarily depended on its knowledge base to provide factual information, but it did not account for health-related changes over time (Ouyang et al., 2025). Some studies combined an LLM with online web searches to identify evidence-based misinformation (Tian et al., 2024; Li et al., 2024a; Dey et al., 2024), which mitigated the limitations of static knowledge base. These systems enhanced evidence retrieval by incorporating information from diverse online sources (Shi et al., 2025). However, web searches introduced new challenges like retrieving low-quality or misleading sources. To address these issues, our study integrated both static and dynamic retrieved evidence to ensure the availability of relevant, real-time information for countering misinformation.

### 2.2 LLM Agents for Social Response Generation

Integrating LLMs across multiple agents has dramatically improved response generation (Zhang et al., 2024; Zhao et al., 2024). Multi-agent LLM systems featured several specialized agents working together to tackle complex tasks, utilizing LLMs to boost the quality and efficiency of interactions (Yang et al., 2025; Guo et al., 2024; Cruz, 2024). Many studies utilized multiple LLM-based multi-agent architectures to improve information retrieval and response generation (Salve et al., 2024), tackling tasks that required long-context understanding (Zhang et al., 2024), and enhancing recommendation quality and user engagement (Fang et al., 2024). For example, Clarke et al. (2022) implemented a framework where agents specialized in different tasks but coordinated to deliver cohesive responses, utilizing a central management system. These multi-agent approaches demonstrated the effectiveness of breaking down complex response generation into modular, specialized components. Inspired by this, our Multi-agent Retrieval-Augmented Framework employed sequentially organized agents, each with distinct capabilities, to retrieve, filter, summarize, generate, and refine evidence-based counterspeech.

---

[5] https://github.com/AnirbanSahaAnik/health-misinformation-reddit-dataset/

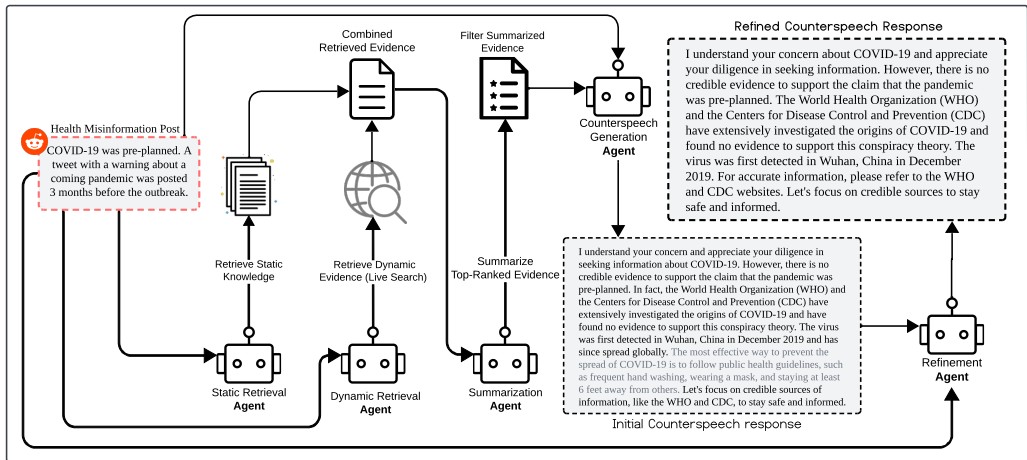

Figure 1: Overview of our proposed Multi-Agent Retrieval-Augmented Framework for addressing health misinformation. First, the Static Retrieval Agent collects evidence from a local database, while the Dynamic Retrieval Agent gathers real-time online information. Next, the Summarization Agent filters and condenses the retrieved evidence for clarity. The Counterspeech Agent generates a response, and the Refinement Agent refines it to ensure clarity, respectfulness, and strong evidential support. This process ensures effective, evidence-based counterspeech.

# 3 Multi-Agent Retrieval-Augmented Framework

## 3.1 Generation

The counterspeech generation process involves leveraging LLMs to create effective, informative, and polite responses that counter health misinformation. The primary objective is to generate responses that not only provide factual corrections but also address misinformation in a respectful and engaging manner. Our proposed multi-agent system uses specialized agents that work in sequence to retrieve, filter, and process evidence, thereby optimizing the RAG pipeline (Figure 1).

**Information Retrieval** To incorporate static and dynamic evidence, two separate agents are employed to retrieve evidence: one is a *Static Retrieval Agent* and another one is a *Dynamic Retrieval Agent*. These Agents collect information based on the health misinformation posts. The Static Retrieval Agent conducts an inquiry within a structured local knowledge base by employing a dual approach that integrates vector-based semantic search with keyword-based exact matching (Yuan et al., 2024; Sawarkar et al., 2024), ensuring that only valid medical knowledge is included in this process. In parallel, the Dynamic Retrieval Agent performs real-time web searches using the DuckDuckGo API, introducing the latest health misinformation corrections from authoritative sources. The two sources of knowledge are combined to produce the final evidence.

**Evidence Summarization** To prevent information overload from combined evidence, we introduce a *Summarization Agent* to filter and distill the most relevant details combined. This step is crucial for eliminating redundancy and resolving conflicting information, ensuring that only the most reliable and pertinent content is retained for counterspeech generation. Specifically, we prompt the Summarization Agent to condense evidence into a clear and refined summary.

**Counterspeech Generation** We integrate distilled evidence from the Summarized Agent into generation by prompting *Counterspeech Generation Agent* to produce evidence-based counterspeech.

**Counterspeech Refinement** To further enhance accessibility and clarity, we employ the *Refinement Agent* to reframe and polish counterspeech, ensuring that the generated counterspeech is not only well-supported by evidence but also contextually relevant and trustworthy.

## 3.2 Evaluation

We employ a comprehensive evaluation framework based on four key criteria: politeness, relevance, informativeness, and factual accuracy. These criteria ensure that the responses are engaging, respectful, factually sound, and effective in addressing health misinformation.

**(a) Politeness:** Politeness in responses is essential in communication, which helps to foster user engagement and build a supportive health community (Shan et al., 2022). A polite counterspeech helps avoid potential backlash and is more likely to be accepted by users (Perez et al., 2025), as it encourages a respectful tone (Song et al., 2025a; Yue et al., 2024; He et al., 2023). To evaluate politeness, we utilize the Multilingual Politeness Classification Model (Srinivasan & Choi, 2022) to assess whether a response adheres to polite discourse norms.

**(b) Relevance:** Relevant responses ensure the response directly addresses the specific health misinformation claim (Yue et al., 2024; He et al., 2023). We assess relevance using the BM25 relevance scoring algorithm (Robertson et al., 2009), which measures how well the generated response aligns with the original health misinformation content.

**(c) Informativeness:** Informativeness reflects the amount of useful, relevant, and evidence-based content shared in the response. A highly informative counterspeech response not only corrects health misinformation but also provides actionable insights and explanations that improve user understanding and support informed decision-making (Chung et al., 2024). To measure informativeness, we employ LLM as an evaluator (Li et al., 2024b) with detailed instructions, assessing if the response provides valuable information (Hong et al., 2025).

**(d) Factual Accuracy:** Factual accuracy demonstrates the reliability and trustworthiness of the generated response by ensuring that the information is correct and supported by evidence (Zhou et al., 2024). In counterspeech generation, presenting scientifically accurate facts can effectively correct misinformation and maintain user trust (Yue et al., 2024). We employ LLM evaluation (Li et al., 2024b) to check how accurate the generated answers are by comparing them to the gathered evidence. The model rates how accurate a response is by checking if it correctly presents scientific facts and does not include any false information.

## 4 EXPERIMENTS AND RESULTS

### 4.1 Dataset

We utilized PRW API [6] to collect health misinformation from Reddit, focusing on topics related to the coronavirus disease (COVID-19), influenza (Flu), and human immunodeficiency virus (HIV), as these topics have been the center of conspiracy theories and misinformation in recent times. We collected Reddit posts and comments containing health-related keywords (e.g., "vaccines," "COVID-19," "alternative medicine", etc. See details in Appendix A.1.2). Finally, we obtained about 4,968 posts and 17,223 comments from high-engagement subreddits (e.g., r/health or r/science, etc. See details in Appendix A.1.1).

After collecting the Reddit dataset, we employed human annotators to identify and filter health misinformation (Song et al., 2025b). Considering the annotation cost, we only extracted 1000 samples for annotation. Five annotators with backgrounds in information science were recruited for the annotation. To ensure consistency and reliability in annotations, all annotators were provided with comprehensive and detailed annotation guidelines Appendix A.2.1. We finally obtained 330 posts containing health misinformation.

---

[6]https://praw.readthedocs.io/

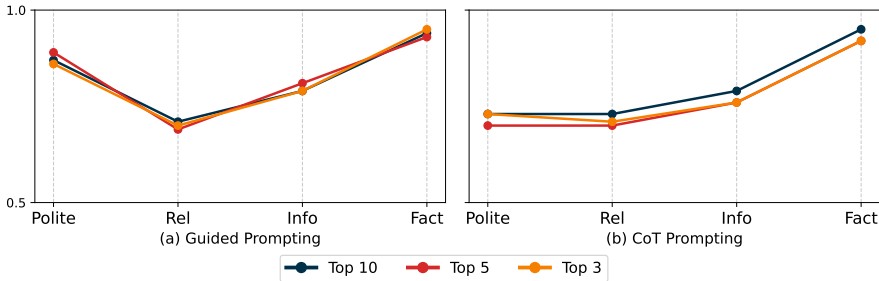

Figure 2: Effect of different top-k evidence filtering on Multi-Agent performance across various metrics (Polite: politeness, Rel: relevance, Info: informativeness, Fact: factual accuracy).

To further enrich our dataset, we developed an automatic classifier trained on the annotated data, enabling the efficient collection of additional health misinformation posts from unannotated data. We employed the RoBERTa-large model (Liu et al., 2019), trained on our manually annotated dataset, which achieves a moderate accuracy with an F1 score of 0.76 (Implementation details in Appendix A.2.2). We applied it to the remaining dataset, identifying 831 additional posts containing health misinformation. To validate the reliability of the classifier, we extracted 100 samples to conduct human validation (See details in Appendix A.2.3). The final dataset contained 1,161 posts labeled as health misinformation.

## 4.2 Baseline

We compared four different baseline methods to determine the most effective way to counter health misinformation against our proposed Multi-Agent Counterspeech Generation System.

**LLM with Direct Prompt (LLM w DP)** This method used direct instructions (Appendix A.4.1) to prompt the LLM to generate counterspeech without any external knowledge. The model took health misinformation posts as input and followed direct instructions to generate responses using the LLM's internal knowledge.

**LLM with Prompt Engineering (LLM w PE)** This method used the refined *Guided Instruction* prompt (Appendix A.4.2) to structure the LLM's responses, ensured clarity, factual accuracy, and coherence while still relying on the LLM's internal knowledge.

**Static RAG** This approach employed RAG to extract supporting evidence from our curated knowledge bases, enabling us to validate the effectiveness of the static evidence in generating accurate and evidence-based responses. The retrieved evidence was then integrated into a *Guided Instruction* prompt (Appendix A.4.2), guiding the LLM in generating evidence-based counterspeech.

**Dynamic RAG** This approach retrieved real-time evidence from the DuckDuckGo search engine, using the misinformation post as a query. By leveraging real-time web search, this method enables dynamic retrieval of the most recent and relevant information, allowing for a comparative analysis between dynamic evidence and static knowledge bases. Notably, this retrieval process followed the same *Guided Instruction* prompt, allowing a comparative assessment of the effectiveness of dynamic evidence in generating counterspeech.

## 4.3 Multi-Agent Retrieval-Augmented Framework Implementation

The Multi-agent Retrieval-Augmented Framework consists of multiple specialized agents that operate sequentially to optimize the retrieval, filtering, summarization, generation, and refinement of responses to health misinformation. We used the *"Llama-3.1-8B-Instruct"* (Grattafiori et al., 2024) model for generation. Generation configurations are detailed in Appendix A.3.

**Knowledge Source** In terms of the static knowledge source, we collected evidence from trusted sources such as the National Institutes of Health (NIH) [7] [8], World Health Organization (WHO) [9] [10] [11]. These institutions are globally acknowledged for their authoritative presence and credibility, offering regular updates on health-related information that are crucial for creating trustworthy counterspeech (Kington et al., 2021). Our dynamic knowledge source utilized the DuckDuckGo search engine API to retrieve real-time, web-based information. This method enables the system to access current evidence from credible news outlets, government health websites, and fact-checking organizations. To empirically assess the potential overlap between the two sources, we computed the BLEU score (Papineni et al., 2002) between retrieved static and dynamic evidence across our samples. The results show a mean BLEU score of 0.0052 with a standard deviation of 0.0033, indicating minimal lexical overlap. This suggests that while dynamic knowledge may occasionally cite static sources, it generally provides distinct and complementary information, supporting the separation used in our system design.

**Static Retrieval Agent** This Agent performed a hybrid search in our curated knowledge base using both Vector-Based Retrieval and Keyword-Based Retrieval. Vector-based retrieval employed a pre-trained embedding model "sentence-transformers/all-mpnet-base-v2" [12] for embedding-based similarity searches and ranking by cosine similarity, while Keyword-Based Retrieval utilized "SimpleKeywordTableIndex" to identify exact team matches within the documents.

**Dynamic Retrieval Agent** The Dynamic Retrieval Agent used the DuckDuckGo API[13] to fetch live external evidence. The misinformation post serves as the structured search query. It extracted key factual sentences from each result via web scraping (using BeautifulSoup[14]). By querying DuckDuckGo, the Dynamic Retrieval Agent can capture the latest posts, news articles, and trusted websites relevant to a given health misinformation post. Additionally, we conducted fact-checking and filtered the information with lower factual accuracy, ensuring that the selected information maintains an average accuracy score above 0.65.

**Summarization Agent** This Agent ranked the retrieved evidence collected from the Static Retrieval Agent and the Dynamic Retrieval Agent. We evaluated various top-k evidence selections, as illustrated in Figure 2, which indicates that top-10 evidence yields a slightly better performance across all dimensions. Therefore, we filtered the top 10 evidence using BM25-based ranking and semantic embedding similarity, and summarized them using the prompt in Appendix A.4.3, effectively preserving key factual details.

**Counterspeech Generation Agent** The summarized evidence was then further utilized by the LLM to generate fact-based counterspeech. This agent employed a structured prompting approach (Appendix A.4.2), which effectively guides the LLM to produce concise, informative, and contextually relevant responses.

**Refinement Agent** The final step involved refining the generated responses using the Refinement Agent. A refined prompt (Appendix A.4.4) guided this process, ensuring that responses are respectful, engaging, and persuasive.

### 4.4 Evaluation Results

Table 1 presents the results of the experiment. The overall performance analysis indicated that LLM **w** DP exhibited slightly strong relevance (0.70) and informativeness (0.77), compared with other baseline methods. Prompt engineering (LLM **w** PE) significantly improved

---

[7]https://www.ncbi.nlm.nih.gov/

[8]https://covid19community.nih.gov/

[9]WHO Misinformation Toolkit

[10]WHO Roles & Responsibilities

[11]WHO FAQ on COVID-19

[12]https://huggingface.co/sentence-transformers/all-mpnet-base-v2

[13]https://pypi.org/project/duckduckgo-search-api/

[14]https://pypi.org/project/beautifulsoup4/

| Method | Politeness | Relevance | Informativeness | Factual Accuracy |
|---|---|---|---|---|
| LLM **w** DP | 0.44 (0.26) | 0.70 (0.14) | 0.77 (0.11) | 0.81 (0.21) |
| LLM **w** PE | 0.84 (0.15) | 0.65 (0.15) | 0.75 (0.07) | 0.83 (0.20) |
| Static RAG | 0.80 (0.15) | 0.65 (0.16) | 0.77 (0.08) | 0.81 (0.24) |
| Dynamic RAG | 0.86 (0.16) | 0.64 (0.15) | 0.73 (0.11) | 0.83 (0.19) |
| **Multi-Agent (Ours)** | **0.88 (0.14)** | **0.70 (0.13)** | **0.78 (0.13)** | **0.86 (0.19)** |

Table 1: Comparison of Multi-Agent counterspeech generation with baseline methods across key evaluation metrics. (DP: direct prompt, and PE: prompt engineering). Higher values indicate better performance. The best scores in each category are underlined. Each score is reported as mean (standard deviation).

politeness (0.84) and factual accuracy (0.83) compared with LLM **w** DP. However, it comes at the cost of slightly lower relevance (0.65).

Introducing static knowledge retrieval (Static RAG) provided better grounding by integrating factual information from a structured knowledge base. However, the results indicated that this did not result in a significant overall improvement across all dimensions compared with LLM without external knowledge (LLM **w** DP and LLM **w** PE), suggesting that LLM may struggle to effectively incorporate external knowledge into generation (Bondarenko & Viehweger, 2024). This pattern remained consistent in the Dynamic RAG experiment (see Figure 3), even when real-time external evidence was provided, further validating that LLMs may have inherent limitations in effectively integrating external knowledge.

Our Multi-agent Retrieval-Augmented Framework successfully addressed these gaps by effectively integrating multiple retrieved evidence and refining the final responses. As a result, it achieved higher politeness (0.88), informativeness (0.78), and factual accuracy (0.86) while maintaining substantial relevance (0.70), making it a more effective and balanced approach to countering misinformation.

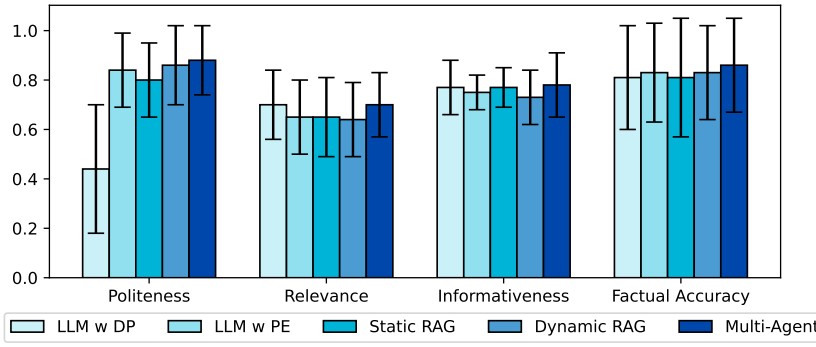

Figure 3: Performance of counterspeech generation approaches across evaluation metrics.

### 4.5 Ablation Experiments

To demonstrate the necessity of multiple agents in our framework, we conducted ablation experiments to generate counterspeech. Additionally, we explored various prompting strategies to assess whether they lead to performance improvements. Table 2 represents how the involvement of different agents contributed to the performance of counterspeech generation under the *Guided* and *CoT* prompting strategies (See in Appendix A.4).

MA **w/o** SA **w/o** RF showed the effect after the removal of Summarization and Refinement Agents. This setup only relied on retrieval and response generation without any additional refinement. It maintained relatively high scores for factual accuracy (Guided: 0.85) and informativeness (Guided: 0.78), however, with lower politeness (Guided: 0.80) compared

| Method | Prompt | Politeness | Relevance | Informativeness | Factual Accuracy |
|---|---|---|---|---|---|
| MA **w/o** SA **w/o** RF | CoT | 0.63 (0.24) | 0.69 (0.13) | 0.76 (0.17) | 0.84 (0.18) |
| MA **w/o** SA **w/o** RF | Guided | 0.80 (0.19) | 0.67 (0.19) | 0.78 (0.13) | 0.85 (0.19) |
| MA **w/o** RF | CoT | 0.65 (0.22) | 0.69 (0.13) | 0.77 (0.09) | 0.86 (0.18) |
| MA **w/o** RF | Guided | 0.79 (0.17) | 0.70 (0.13) | 0.82 (0.15) | 0.87 (0.19) |
| MA | CoT | 0.74 (0.19) | 0.67 (0.14) | 0.77 (0.09) | 0.87 (0.19) |
| **MA (Ours)** | **Guided** | **0.88 (0.14)** | **0.70 (0.13)** | **0.78 (0.13)** | **0.86 (0.19)** |

Table 2: Evaluation of the impact of Summarization Agent (SA) and Refinement Agent (RF) alongside different prompting styles (Guided vs. CoT) on Multi-Agent (MA) counterspeech generation performance. Each score is reported as mean (standard deviation).

with MA. Including the Summarization Agent (MA **w/o** RF with *Guided*) improved the relevance (0.70), factual accuracy (0.87), and informativeness (0.82) compared with MA **w/o** SA **w/o** RF. This further demonstrated that summarized evidence helped LLMs better integrate evidence into generation. Furthermore, the addition of the Refinement Agent, our proposed Multi-Agent approach (MA (Ours)), further refined responses by ensuring they were organized, polite, and clear, with the best performance across all dimensions.

We also examined the effect of changing the Prompting Strategy to Chain-of-Thought (CoT) (See details in Appendix A.4.5). While the CoT prompting strategy enabled the generation to maintained similar factual accuracy compared with Guided prompting, the performance of politeness (MA **w/o** SA **w/o** RF: 0.63 vs 0.80; MA **w/o** RF: 0.65 vs 0.79; MA: 0.74 vs 0.88), and informativeness (MA **w/o** SA **w/o** RF: 0.76 vs 0.78; MA **w/o** RF: 0.77 vs 0.82; MA: 0.77 vs 0.78) decreased. The results indicated that CoT reasoning may be less effective than Guided prompting in our study. The step-by-step reasoning process often introduced redundant or overly detailed content, which decreased politeness and informativeness (Sprague et al., 2025; Wei et al., 2022).

## 4.6 Cross Evaluation

**Topic-wise Evaluation** To further assess the robustness of our framework across different types of misinformation, we conducted a topic-wise analysis covering COVID-19, HIV, and Influenza. As shown in Appendix A.5 (Table 7), our Multi-Agent framework consistently achieved strong average scores across all topics and evaluation dimensions, with the following scores: politeness (0.90), relevance (0.71), informativeness (0.77), and factual accuracy (0.85). This further demonstrated its generalizability across diverse health misinformation domains.

| Method | Politeness | Relevance | Informativeness | Factual Accuracy |
|---|---|---|---|---|
| LLM **w** DP | 0.44 (0.26) | 0.70 (0.14) | 0.77 (0.11) | 0.81 (0.21) |
| LLM **w** PE | 0.81 (0.18) | 0.77 (0.11) | 0.77 (0.07) | 0.94 (0.12) |
| Static RAG | 0.86 (0.14) | 0.77 (0.12) | 0.78 (0.09) | 0.95 (0.12) |
| Dynamic RAG | 0.85 (0.16) | 0.77 (0.13) | 0.76 (0.10) | 0.92 (0.16) |
| **Multi-Agent (Ours)** | **0.89 (0.11)** | **0.78 (0.10)** | **0.83 (0.12)** | **0.96 (0.12)** |

Table 3: Cross-platform generalization results on the MisinfoCorrect dataset (He et al., 2023). Our method achieves the highest performance across all evaluation metrics. (DP: direct prompt, and PE: prompt engineering). Each score is reported as mean (standard deviation).

**Cross-Platform Generalization** We evaluated our system on the external MisinfoCorrect dataset (He et al., 2023), which contains COVID-19 vaccine misinformation from Twitter (now X). This experiment tests cross-platform generalization. Results, shown in Table 3, demonstrated that our method maintained top performance on this cross-platform dataset, with politeness (0.89), relevance (0.78), informativeness (0.83), and factual accuracy (0.96).

## 5 Human Evaluation

We conducted multiple rounds of human evaluations to assess the quality and effectiveness of the generated counterspeech comprehensively. The first round was to validate the LLM evaluation. Even though various studies relied on LLMs as evaluators (Li et al., 2024b; Evans et al., 2024; Chen et al., 2024), different research tasks could yield varying LLM performance (Chern et al., 2024; Chan et al., 2024). Therefore, we involved humans to verify the LLM's evaluation of informativeness and factual accuracy. The second round aimed to explore the effectiveness of the refined response. In our study, we utilized the Refinement Agent to improve counterspeech; however, it remained unknown whether users preferred these refined responses. Therefore, it was necessary to determine human preference, which might help us improve generation in future studies. In the final round of human evaluation, we assessed the overall quality of counterspeech across systems. We compared responses generated by our proposed Multi-Agent framework against two strong baselines, Static RAG and Dynamic RAG, based on human judgments of effectiveness in correcting health misinformation.

In the first stage, we conducted a human evaluation to verify the reliability of the computational evaluation, focusing on informativeness and factual accuracy (details are provided in Appendix A.6). For factual accuracy, Cohen's Kappa was $\kappa \geq 0.69$, and for informativeness, Cohen's Kappa was $\kappa \geq 0.65$, which indicated substantial agreement with the model evaluation. These results suggested that the computing-based evaluation exhibited strong consistency with human judgments.

Secondly, we assessed the effectiveness of the Refinement Agent in counterspeech responses (details of the guidelines in Appendix A.7). The initial agreement rates among them were 74%, 71%, and 73%, respectively, which showed they had a relatively high level of consistency. The annotation results showed that 72% of users preferred the responses generated with the Refinement Agent, which indicated that clearer and better-organized responses are more accessible for users. In the future, we could continue refining our methods to produce counterspeech that is even more user-friendly and effective.

In the final round, we compared the overall effectiveness of our proposed Multi-Agent Framework with two strong baselines, Static RAG and Dynamic RAG (detailed in Appendix A.8). The inter-annotator agreement was substantial, with an agreement rate of 80% and a Cohen's Kappa of $\kappa \geq 0.69$. Results showed that the counterspeech generated by our proposed framework was selected as the most effective in 47% of cases, compared to 32% for Static RAG and 21% for Dynamic RAG. These findings suggested that our method achieved superior overall quality based on human evaluations.

## 6 CONCLUSION

In this study, we presented a Multi-Agent Retrieval-Augmented Framework for generating effective counterspeech against health misinformation. This framework integrated multiple specialized LLM agents to sequentially (1) retrieve evidence from both static knowledge bases and dynamic web sources, (2) filter and summarize the combined evidence, and (3) generate and refine counterspeech using structured prompting. This design enables the generation of responses that were not only factually accurate but also polite, informative, and contextually relevant. Experimental results and human evaluations demonstrated that our method outperforms several baselines across multiple dimensions. Additionally, ablation studies confirmed the importance of each module, particularly the refinement agent, which played a key role in enhancing the clarity and engagement of counterspeech. Our findings highlighted the value of combining static and dynamic knowledge and the refined method to improve the reliability and quality of automated counterspeech. We acknowledge several limitations in our study. We recognize that we haven't fully collected all existing static knowledge due to the cost of manual work, and our system may require many computing resources. In future work, we aim to further refine the curation processes for knowledge sources and explore methods to improve the efficiency of our framework.

## Ethics Statement

Our study follows the highest ethical standards to address misinformation using Large Language Models. We acknowledge that AI-generated responses can affect society, and we are careful to ensure they are effective while still being ethical. We use publicly available data sources, ensuring that no personally identifiable information is collected or used. User identities from social media datasets are anonymized, and any sensitive information is removed before analysis. Human annotators involved in data annotation and evaluating counterspeech responses remain anonymous, and their participation is voluntary. Annotators are compensated on average with $15 per hour. We follow rules to make sure the annotation process is fair and open. Our study aims to generate respectful, fact-based counterspeech while minimizing unintended biases. We understand that LLMs can show biases based on the data on which they were trained. To address this, we use retrieval-augmented methods to ensure our answers are based on verified facts. We ensure that the AI-generated responses are not misleading, confrontational, or harmful. The system is created to give polite and well-structured responses that encourage helpful discussion rather than increasing division. We are committed to transparency in our methodology, which includes providing details on model architectures, data sources, and evaluation criteria. Our work aligns with the principles of ethical AI research and aims to positively contribute to public discourse by countering misinformation with reliable evidence. By following these ethical considerations, we aim to create AI solutions that encourage good communication, build public trust, and deliver accurate, respectful, and reliable information.

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

# A  Appendix

## A.1   Subreddits and Keyword lists

### A.1.1   List of Subreddits used

Table 4 presents the list of subreddits analyzed in this study. These subreddits were selected to cover a broad spectrum of discussions on science, public health, medical topics, vaccines, and conspiracy theories.

| Subreddit Name |
| --- |
| r/science, r/Wuhan_Flu, r/CoronavirusCirclejerk, r/DebateVaccines, r/vaccinelonghaulers, r/ChurchofCOVID, r/vaccineautismevidence, r/ScienceUncensored, r/CoronavirusUS, r/conspiracy, r/medical_advice, r/joerogan, r/Conservative, r/vaccineaddiction, r/thingsprovaxxerssay |

Table 4: List of subreddits analyzed in the study.

### A.1.2   List of keywords used

Table 5 presents the list of keywords used to extract posts and submissions from the selected subreddits. These keywords encompass a wide range of medical, health, and pandemic-related discussions.

| Keywords |
| --- |
| diagnosis, treatment, prescription, prognosis, nutrition, diet, exercise, chemicals, abortion, vaccine, jabs, COVID-19, surgery, pills, symptom, immunity, disease, autism, cure, infection, infected, pandemic, health, Corona, virus, medical, doctor, insomnia, side effects, polio, symptoms, fibromyalgia, acidity, cancer, tuberculosis, HIV, medication, contraception, immunocompromised, covid, vax, truth, sick, flu, pregnancy, pneumonia, measles, jab. |

Table 5: List of keywords used to extract relevant posts from subreddits.

## A.2   Guidelines and Annotation process for detecting health misinformation

### A.2.1   Annotation guidelines and process

After collecting the Reddit dataset, the next step was to annotate posts into two categories: "Health Misinformation" and "Not Health Misinformation.". To ensure annotation consistency, all annotators received comprehensive guidelines, including examples of health misinformation, classification criteria, and citations of reputable sources (e.g., CDC, WHO, and NIH) for verification purposes.

**Annotation guidelines for health misinformation labeling**

```
The goal of this annotation task is to classify posts based on whether they
contain health-related misinformation. Posts will be assigned one of two labels:

Label 1: "Health Misinformation", if the post contains health-related misinformation.
Label 0: "Not Health Misinformation", if the post does not contain any
health-related misinformation or is unrelated to health information.

Definition of Health Misinformation:
Any false, misleading, or unverified claim related to health, medicine,
diseases, treatments, vaccines, nutrition, or wellness.
Misinformation includes claims that contradict established medical research, public
health guidelines, or authoritative sources such as the World Health Organization
(WHO), the Centers for Disease Control and Prevention (CDC), and the National
Institutes of Health (NIH).
```

**Annotation process**

The annotation process was conducted in two stages to ensure consistency and accuracy. In the first stage, three annotators independently labeled 480 posts as either health misinformation or not health misinformation. In the second stage, another set of three annotators annotated 499 additional posts using the same classification scheme. Each annotator independently reviewed the content and fact-checked it against authoritative sources before assigning labels.

Despite the structured guidelines, some posts posed challenges in classification, particularly those containing satirical statements, ambiguous claims, or partially correct yet misleading information. In the first stage, the Cohen's Kappa agreement rates between annotators were $\kappa \geq 0.58$, $\kappa \geq 0.67$, and $\kappa \geq 0.54$, indicating moderate to substantial agreement. In the second stage, the agreement rates were $\kappa \geq 0.69$, $\kappa \geq 0.48$, and $\kappa \geq 0.56$, also reflecting a mix of moderate and substantial agreement. To resolve disagreements in the annotations, the annotators participated in group discussions to ensure the final classification was well-supported by evidence. In cases where uncertainties persisted, the posts were cross-checked against fact-checking websites such as Snopes, HealthFeedback, and FactCheck.org to verify the accuracy of the information. This multi-step validation process ensured that the final dataset maintained a high level of accuracy and reliability.

After resolving all disagreements and finalizing the labels, the dataset consisted of 330 posts labeled as health misinformation and 648 posts labeled as not health misinformation.

### A.2.2 RoBERTa-large Implementation Details

We fine-tuned the `roberta-large`[15] model to classify Reddit posts as health misinformation or not, using the post as input. The model was trained on a class-balanced dataset with a maximum sequence length of 128, a batch size of 16, and a learning rate of $2e^{-5}$ for 4 epochs. We used the Adam optimizer and evaluated performance on a 20% test set.

Table 6 summarizes the classifier's performance across both classes. The model achieved a weighted F1-score of 0.76.

| Class | Precision | Recall | F1-Score |
|---|---|---|---|
| Not Health Misinformation | 0.71 | 0.84 | 0.77 |
| Health Misinformation | 0.82 | 0.68 | 0.74 |
| **Weighted Average** | **0.77** | **0.76** | **0.76** |

Table 6: Performance of the RoBERTa-large classifier on the human-annotated Reddit test set.

### A.2.3 Human validation

A human validation step was conducted to validate the reliability of these automatically labeled posts. Three annotators independently fact-checked a random sample of 100 model-labeled posts, assessing their accuracy. The agreement rate between the annotators is 88.1%, 89.1%, and 85.1%, and Cohen's Kappa scores are $\kappa \geq 0.73$, $\kappa \geq 0.75$, and $\kappa \geq 0.67$, respectively. Which shows substantial agreement between the annotators. For the disagreement, we conducted further discussion and fact-checking, which concluded us with the final label. With the final label, our model agreement rate is 88.1% and Cohen's Kappa, $\kappa \geq 0.73$ demonstrates a substantial agreement between the model's classification and human judgment.

---

[15]https://huggingface.co/FacebookAI/roberta-large

### A.3 Computing resources and model parameters

#### A.3.1 Hardware configuration

We utilized a multi-GPU setup with the following specifications:

- **CPU:** Intel® Xeon® Gold 6226R @ 2.90GHz
    - Cores: 16 per socket, total 32 cores
    - Threads: 64 (2 threads per core)
- **Memory:** 125GB RAM
- **GPU:** Three NVIDIA Quadro RTX 8000 GPUs
    - CUDA Version: 12.6
    - Memory: 48GB per GPU (Total 144GB VRAM)

Our Multi-agent Retrieval-Augmented Framework was deployed across three GPUs, leveraging parallel processing for retrieval, summarization, and response generation. To ensure efficiency, data pre-processing was optimized through multithreading, and retrieval mechanisms combined keyword-based and embedding-based search strategies.

#### A.3.2 Model and hyperparameter settings

The key hyperparameter settings for all experiments are as follows:

- **LLM:** Meta-Llama 3.1-8B-Instruct[16]
- **Embedding Model:** Sentence-Transformers (all-mpnet-base-v2)[17]
- **Summarization:**
    - Temperature: 0.3
    - Top-p: 0.8
- **Counterspeech Generation:**
    - Temperature: 0.6
    - Top-p: 0.85
- **Response Refinement:**
    - Temperature: 0.4
    - Top-p: 0.85

### A.4 Prompt Design

#### A.4.1 Direct instructions prompt for counterspeech generation:

Generate a counterspeech response to the given health misinformation post:

Post: {post_content}

Counterspeech:

#### A.4.2 Prompt Engineering (Guided prompt) for counterspeech generation:

**Guided prompt:**

You are an AI designed to provide concise, respectful, and evidence-based counterspeech to health misinformation in the form of a comment responding directly to a post.

---

[16]https://huggingface.co/meta-llama/Llama-3.1-8B-Instruct
[17]https://huggingface.co/sentence-transformers/all-mpnet-base-v2

```
Guidelines:
- Begin with an empathetic tone to validate user concerns before addressing
  the misinformation.
- Clearly and succinctly correct misinformation using simple, factual
  explanations.
- Integrate concise evidence from trusted sources (e.g., CDC, WHO, NIH) with
  clear attribution, limiting to 2-3 references.
- Avoid redundant information, unrelated anecdotes, or excessive links.
- Craft responses that are precise, easy to understand, and actionable,
  keeping them under 120 words in a direct reply format.
```

**Counterspeech generation prompt (static & dynamic):**

```
{Guided Prompt}

Post: {post_content}

Evidence: {evidence}

Response:
```

**Counterspeech generation prompt (Multi-Agent):**

```
{Guided Prompt}

Misinformation Post:
{post_content}

Verified Evidence:
{summarized_evidence}

Write a counterspeech response that:
- Directly refutes the misinformation in a confident yet respectful manner.
- Provides an example that explains why the misinformation is incorrect.

counterspeech Response:
```

### A.4.3   Summarization Agent prompt

```
Below is a collection of factual evidence addressing misinformation:

{combined_filter_evidence}

Summarize this evidence concisely, preserving key factual
details while removing redundant or speculative information.

Final Summarized Evidence:
```

### A.4.4   Refinement Agent prompt

```
Below is a counterspeech response to a misinformation Reddit post.
Refine the response to ensure it is clear, factually accurate, and
respectful, without reducing its effectiveness in addressing the misinformation.

Misinformation Post:
{post_content}

Current counterspeech Response:
```

```
{response}
```

Refinement Guidelines:
- Begin with an empathetic and respectful tone.
- Be more polite, respectful while clearly addressing the misinformation.
- Improve clarity and readability while keeping key factual evidence intact.
- Ensure the final response is factually accurate without removing or altering
any factual information.
- Add references from trusted sources if available.
- Keep the response concise (under 120 words), but make sure it does not
lose relevance.

Refined Response:

### A.4.5 Chain-of-Thought (CoT) prompt for generation

**Chain-of-Thought (CoT) counterspeech generation prompt:**

You are an AI fact-checking specialist designed to analyze health
misinformation and generate counterspeech through structured reasoning.
Your process must:
1. Maintain scientific accuracy while being accessible
2. Prioritize harm reduction in responses
3. Balance empathy with factual rigor
4. Adhere to medical consensus from WHO/CDC/NIH

Health Misinformation Post:
{post_content}

Verified Evidence:
{summarized_evidence}

Generate a counterspeech response to this Health Misinformation Post
using the following reasoning process:

1. Analyze the post to identify key claims and emotional undertones.
2. Select the most relevant facts from verified evidence.
3. Explain why the claims are misleading using:
   a) Logical contradictions in the claims.
   b) Evidence from authoritative sources.
   c) Concrete examples/counter-examples.
4. Structure the response with:
   - Empathetic acknowledgement.
   - Clear factual correction.
   - Simple analogy/metaphor for understanding.
   - Keep the response concise, under 150 words.
   - Trusted source references.

Reasoning Process:
[Think through each step carefully]

counterspeech Response:

**Chain-of-Thought (CoT) prompt for counterspeech refinement:**

Refine the generated counterspeech response to a health misinformation post
while maintaining clarity, factual accuracy, and politeness.

1. Identify and resolve logical contradictions.
2. Strengthen arguments with source-attributed evidence.

```
3. Improve structure for effective communication (acknowledgment → correction
→ analogy).

Misinformation Post:
{post_content}

Current counterspeech Response:
{response}

Refinement Guidelines:
- Rewrite with a respectful and empathetic tone.
- Ensure analogies/examples are universally understandable.
- Remove redundant explanations while ensuring clarity.
- Trim to 120 words without losing key evidence.

Output Format:
[Empathetic and Respectful Opening]
[Clear Factual Correction]
[Simple Analogy/Example]
[Trusted Source References]

Refined Response:
```

## A.5 Evaluation Across Topics

Our dataset covers a range of health misinformation topics, including COVID-19, HIV, and Influenza. To assess the robustness of our approach across domains, we aggregated evaluation results by topic. As shown in Table 7, our Multi-Agent framework achieves consistently strong average performance across all three topics.

| Method | Category | Politeness | Relevance | Informativeness | Factual Accuracy |
|---|---|---|---|---|---|
| LLM w DP | COVID-19 | 0.45 (0.27) | 0.74 (0.17) | 0.75 (0.14) | 0.78 (0.21) |
| | HIV | 0.57 (0.31) | 0.78 (0.12) | 0.76 (0.06) | 0.75 (0.27) |
| | Influenza | 0.42 (0.20) | 0.81 (0.10) | 0.75 (0.08) | 0.70 (0.24) |
| | **Average** | **0.48 (0.26)** | **0.78 (0.13)** | **0.75 (0.09)** | **0.74 (0.24)** |
| LLM w PE | COVID-19 | 0.78 (0.20) | 0.69 (0.12) | 0.76 (0.06) | 0.81 (0.24) |
| | HIV | 0.83 (0.17) | 0.71 (0.12) | 0.75 (0.00) | 0.79 (0.17) |
| | Influenza | 0.86 (0.13) | 0.73 (0.08) | 0.76 (0.06) | 0.71 (0.25) |
| | **Average** | **0.82 (0.17)** | **0.71 (0.11)** | **0.76 (0.04)** | **0.77 (0.22)** |
| Static RAG | COVID-19 | 0.75 (0.18) | 0.67 (0.12) | 0.78 (0.08) | 0.89 (0.17) |
| | HIV | 0.81 (0.13) | 0.70 (0.17) | 0.76 (0.06) | 0.68 (0.35) |
| | Influenza | 0.75 (0.16) | 0.73 (0.10) | 0.76 (0.06) | 0.75 (0.26) |
| | **Average** | **0.77 (0.16)** | **0.70 (0.13)** | **0.77 (0.07)** | **0.77 (0.26)** |
| Dynamic RAG | COVID-19 | 0.84 (0.23) | 0.62 (0.20) | 0.70 (0.10) | 0.88 (0.19) |
| | HIV | 0.91 (0.12) | 0.70 (0.14) | 0.73 (0.08) | 0.84 (0.27) |
| | Influenza | 0.82 (0.23) | 0.69 (0.12) | 0.74 (0.10) | 0.66 (0.26) |
| | **Average** | **0.86 (0.19)** | **0.67 (0.15)** | **0.72 (0.09)** | **0.79 (0.24)** |
| **Multi-Agent (Ours)** | COVID-19 | 0.92 (0.05) | 0.68 (0.17) | 0.78 (0.08) | 0.84 (0.19) |
| | HIV | 0.86 (0.14) | 0.73 (0.09) | 0.78 (0.08) | 0.91 (0.17) |
| | Influenza | 0.93 (0.14) | 0.71 (0.10) | 0.74 (0.06) | 0.79 (0.15) |
| | **Average** | **0.90 (0.11)** | **0.71 (0.12)** | **0.77 (0.07)** | **0.85 (0.17)** |

Table 7: Topic-wise evaluation results across all systems for COVID-19, HIV, and Influenza. Standard deviation is reported in parentheses.

### A.6 Annotation guidelines for factual accuracy and informativeness assessment

The primary objective is to determine how well the model-generated evaluation scores align with human judgments. We randomly select 100 samples, and three annotators independently evaluate the generated responses based on informativeness and factual accuracy on a 1-5 scale. Initially, the broad rating scale results in variability in annotator agreement. To address this, further discussions among the annotators are held, leading to a final label that reflects a consensus-driven evaluation. Once the final human labels are established, we compared them with our model evaluation results.

**Guideline for informativeness (1-5 Scale) :**

```
Evaluate the informativeness of the following counterspeech response on a scale
of 1 to 5.

Score 1: Not Informative: The response lacks substance or is irrelevant.
Score 2: Slightly Informative: Some relevant points but too vague or missing
key details.
Score 3: Moderately Informative: Provides a basic understanding but lacks
depth.
Score 4: Mostly Informative: Covers most aspects well, with minor gaps.
Score 5: Completely Informative: Comprehensive, well-structured, detailed,
and highly informative.
```

**Factual accuracy evaluation guidelines (1-5 Scale):**

```
Your task is to assess the factual accuracy of the counterspeech response using
the provided evidence. Compare the response against the evidence and determine
its correctness using the scoring criteria below. You need to verify the
link/reference (if available) in the counterspeech response and also need to
write a comment explaining why you chose this score.

Score 1: Completely Inaccurate: Mostly inaccurate, contains false claims.
Score 2: Mostly Inaccurate: Some correct facts, but also contains misleading
information.
Score 3: Partially Accurate: Mostly correct, but lacks sufficient evidence.
Score 4: Mostly Accurate: Accurate, well-supported, minor inconsistencies.
Score 5: Completely Accurate: Fully fact-based, verified by reliable sources.
```

### A.7 Human evaluation guidelines for counterspeech comparison

We conduct a comparative human evaluation of two different response strategies: Response A (Multi-Agent **w/o** Refinement Agent) and Response B (Multi-Agent **w** Refinement Agent). The goal is to determine human preference and assess how the Refinement Agent enhances response quality.

```
Your task is to compare two responses (Response A and Response B) that address
a misinformation post.
You will decide which response is better based on your preference.

How to Evaluate:

1. Read the Misinformation Post carefully to understand the claim being made.
2. Compare Response A and Response B.
3. Select the response you may prefer to use in addressing the misinformation.
4. Provide a brief comment explaining why you selected that response.

Note: Response A (Multi-Agent w/o Refinement Agent) and
      Response B (Multi-Agent w Refinement Agent)
```

### A.8 Overall Effectiveness Evaluation

To assess the overall quality of responses in correcting misinformation, we conducted a human evaluation comparing three systems: our Multi-Agent framework (Response A), Static RAG (Response B), and Dynamic RAG (Response C).

We asked three annotators to independently review 100 sampled misinformation posts along with the counterspeech responses generated by each system. Annotators were instructed to select the response that was most effective overall. Effectiveness was defined as how well the response refutes the misinformation and whether it is persuasive, credible, and well-structured.

**Annotation Guideline**

```
Your task is to compare three responses (Response A, Response B, and Response C)
that address a specific misinformation post. You will assess which response is
more effective based on the criteria below.

Evaluation Criteria:

1. Clarity: Is the response easy to understand? Does it avoid technical jargon
or overly complex language?
2. Politeness: Is the tone respectful and considerate? Does it correct
misinformation without sounding rude or confrontational?
3. Factual Accuracy: Are the facts correct and supported by evidence?
Does the response reference or align with reliable sources?
4. Informativeness: Does the response provide useful explanations, context,
or next steps? Is it educational and helpful in promoting understanding?
5. Overall Effectiveness: Does the response effectively refute misinformation?
Is it persuasive, credible, and well-structured?

How to Annotate:

1. Carefully read the Misinformation Post to understand the claim.
2. Compare Response A, Response B and Response C) across all five dimensions.
3. Choose the response that is more effective overall in correcting misinformation.
```

