# OpenReview forum: "Multi-Agent Retrieval-Augmented Framework for Evidence-Based Counterspeech Against Health Misinformation"
_colmweb.org/COLM/2025/Conference — COLM 2025_

### Official Review · Reviewer_7B6M · 2025-05-11

**Rating:** 7
**Confidence:** 4
**Ethics Flag:** 1

**Summary:**

This paper proposes a new Multi-Agent Retrieval-Augmented Framework for generating counterspeech against health misinformation. The core idea is to leverage both static knowledge (verified information that is relatively stable and rarely changes) and dynamic knowledge (information that is continually updated and responsive to evolving situations). The framework specifically aims to generate polite counterspeech and incorporates several agents, including a summarizer. A new dataset is sampled from social media posts and comments, where human annotators identify posts containing misinformation. These posts serve as input to the proposed framework and four baseline models. The outputs of each system are automatically evaluated across multiple quality dimensions, including politeness, relevance, informativeness, and factual accuracy. Results show that the proposed framework outperforms the baselines on all dimensions. A small manual study is conducted to validate the automatic evaluations of Informativeness and Factual Accuracy, showing good correlation with human judgments.

**Questions To Authors:**

- Did you control for the length of the generated counterspeech across the systems? I suspect that differences in length could influence the evaluations of informativeness, relevance, and politeness.
- The politeness score of the LLM w DP seems low; do you have any insights into why this might be the case, especially considering that LLMs are typically optimized for polite language, as far as I know?
- Some related work:

https://aclanthology.org/2024.acl-long.702.pdf

https://arxiv.org/pdf/2502.13847

https://arxiv.org/pdf/2406.07348

**Reasons To Accept:**

- The idea of combining both static and dynamic knowledge in a retrieval-augmented framework is conceptually sound. While not entirely novel. given recent related work, it is applied here to a new and important task (counterspeech against health misinformation), which makes it a nice contribution. This level of novelty, in my opinion, is appropriate for COLM.
- The design choices made in the framework are well-reasoned and clearly justified. The overall system architecture, including the use of multiple agents and the integration of both knowledge types, seems reasonable. The evaluation setup also appears to be sound and covers relevant quality dimensions.
- The proposed framework demonstrates strong performance, clearly outperforming four strong baselines across multiple evaluation metrics.

**Reasons To Reject:**

- Some implementation details are missing. For example, the paper does not provide information about the RoBERTa-large model used for misinformation classification, which is an important step in data preparation.
- The dataset is somewhat limited in diversity, as it focuses solely on social media posts covering a narrow range of health-related misinformation topics. I think incorporating a broader range of misinformation types and sources would improve generalizability.
- Although I appreciate the human validation study for evaluating informativeness and factual accuracy, a more informative evaluation, in my opinion, could have involved human comparisons of full outputs from the proposed framework, Static RAG, and Dynamic RAG.

---

> ### Author Response · Authors · 2025-06-02
>
> ### **W1:** Implementation Details about RoBERTa-large model
>
> **R1:** Thanks for your suggestion. We fine-tuned the RoBERTa-large model from Hugging Face Transformers to classify Reddit posts as health misinformation or not, using the post titles as input. The model was trained on a class-balanced dataset with a maximum sequence length of 128, a batch size of 16, and a learning rate of 2e-5 for 4 epochs. We used the Adam optimizer and evaluated performance on a 20% test set. The following table summarizes the classifier's performance across both classes. The model achieved a weighted F1-score of 0.76. We will clarify it in our revised version. We emphasize that the model is used for the initial identification of posts likely to contain health misinformation. Human annotators then review, annotate, and discuss these posts to finalize the selection of misinformation content for the counterspeech experiments.
>
> | **Class**                 | **Precision** | **Recall** | **F1-Score** |
> | ------------------------- | ------------- | ---------- | ------------ |
> | Not Misinformation (0)    | 0.71          | 0.84       | 0.77         |
> | Health Misinformation (1) | 0.82          | 0.68       | 0.74         |
> | **Weighted Average**      | 0.77          | 0.76       | 0.76         |
>
>
>
> ### **W2:** Human comparisons of full outputs from the proposed framework, Static RAG, and Dynamic RAG
>
> **R2:** Thank you for your valuable insights. We have conducted additional human evaluations to assess the overall quality of counterspeech generated by our proposed framework, Static RAG, and Dynamic RAG.
>
> We asked three annotators to independently review 100 sampled misinformation posts along with the responses generated by our proposed framework, Static RAG, and Dynamic RAG, and to select the response that was most effective overall in correcting the misinformation. All three annotators hold master’s degrees and have academic backgrounds in information science. They also have prior experience in evaluating online discourse and content moderation, making them well-qualified to assess the quality and effectiveness of counterspeech responses. Overall effectiveness is defined as how well the response refutes the misinformation, and whether it is persuasive, credible, and well-structured. The inter-annotator agreement is substantial, with an agreement rate of 0.80 and a Cohen’s Kappa of 0.69.
>
> Results show that the counterspeech generated by our proposed framework is selected as the most effective in 47% of cases, compared to 32% for Static RAG and 21% for Dynamic RAG. These findings suggest that our method achieves superior overall quality based on human evaluations. We will add this evaluation to the revised manuscript.
>
>
> ### **W3:** The length of the generated counterspeech
>
> **R3:** Thank you for this insightful question. All systems, including both the baselines and our proposed framework, are prompted with explicit instructions to generate responses under 120 words. This constraint is designed not only to ensure fair comparability across systems, but also to align with user preferences, shorter responses are often more persuasive, accessible, and effective, especially in online discourse settings [(Braca & Dondio, 2023)](http://dx.doi.org/10.1007/s43039-023-00077-0). We will clarify this in the revised manuscript.
>
>
>
> ### **W4:** Why the politeness score of the LLM w DP seems low considering that LLMs are typically optimized for polite language?
>
> **R4:** Thank you for this thoughtful observation. While it is true that instruction-tuned LLMs are generally optimized for polite and safe language, we believe the lower politeness score in the LLM w DP setting reflects the limited ability of LLMs to consistently produce polite responses without explicit guidance. Our Prompting experiment explicitly instructs the model to begin with empathetic language and maintain a respectful tone throughout. This structured prompting significantly improves the politeness of responses, as supported by our evaluation results. We will clarify this point in the revised manuscript and emphasize the role of prompt design in shaping stylistic qualities such as politeness.
>
>
>  **Reference:**
>
> Braca, A., & Dondio, P. (2023). *Developing persuasive systems for marketing: The interplay of persuasion techniques, customer traits and persuasive message design*. *Italian Journal of Marketing, 2023*(3), 369–412. [http://dx.doi.org/10.1007/s43039-023-00077-0](http://dx.doi.org/10.1007/s43039-023-00077-0)

---

> > ### Comment · Reviewer_7B6M · 2025-06-09
> >
> > Thank you for the reply. I will keep my score as it is.

---

### Official Review · Reviewer_dAts · 2025-05-12

**Rating:** 5
**Confidence:** 3
**Ethics Flag:** 1

**Summary:**

This paper proposes a multi-agent framework that retrieves both static and dynamic knowledge to generate potentially more factually reliable counterspeech. Overall, the paper is quite easy to follow; the description of each agent and how they collaborate within the framework is clearly presented. The evaluation results show that the proposed framework indeed outperforms the baselines. However, the paper does not sound exciting enough to me and definitely needs more justification regarding the details of the framework.

**Reasons To Accept:**

1. This paper is easy to follow.
2. The human annotated data used in this paper could be helpful for later research if open-sourced.

**Reasons To Reject:**

1. The novelty of this paper appears to be somewhat limited. The idea of combining both static and dynamic information for generating counterspeech is relatively straightforward and intuitive. Furthermore, although the authors claim that prior work only leverages LLMs with online sources and does not incorporate static databases, LLMs themselves can be considered as a form of static knowledge base, where information is encoded in their parameters and can be retrieved in ways analogous to querying a traditional database.
2. I think the authors should provide more explanations regarding the design of the proposed framework. In particular, it would be beneficial to explore alternative configurations or task choices for each agent and how these components interact and collaborate within the overall system.  I am also curious whether the use of this specific number of agents is essential to the framework, or if a smaller number of agents could achieve comparable results.
3. Last, the performance gain of the proposed framework is limited considering how much resource is put into using this framework.

---

> ### Author Response · Authors · 2025-06-02
>
> ### **W1:** The novelty of this paper appears to be somewhat limited.
>
> **R1:** Thank you for your comment. While our approach builds on existing techniques, we believe our contribution lies in the following aspects, which have intellectual merits and practical implications.
>
> **(a) Integration of Static and Dynamic Knowledge:**
> The framework combines both static (e.g., curated medical guidelines) and dynamic (e.g., real-time web search) evidence to generate counterspeech, enabling responses that are both authoritative and up to date. To our knowledge, this combination has not been explored before, especially in the generation of counterspeech to health misinformation. Our experiments show the significance of including both static and dynamic knowledge.
>
>
>
> **(b) Modular Multi-Agent Pipeline Specialized for Counterspeech Generation Against Health Misinformation:**
> This pipeline presents a modular multi-agent framework tailored for generating effective counterspeech in response to health misinformation. While individual components such as RAG have been studied before, our contribution lies in the integration of dedicated sub-agents responsible for:
> (1) static and dynamic retrieval,
> (2) evidence summarization,
> (3) counterspeech generation, and
> (4) iterative response refinement.
>
> Each agent functions autonomously yet collaboratively within a scalable architecture, enhancing the quality of the generated counterspeech. Experiments show each agent contributes to the improved quality, relevance, and factual grounding in the generated counterspeech. The structured pipeline with these four agents has not been previously applied or evaluated for the generation of counterspeech to health misinformation.
>
>
> **(c) Ablation Studies with Prompting Strategy Comparison:**
> Comprehensive ablation studies evaluate the contribution of each agent in the pipeline. The system is tested under different prompting strategies (e.g., Guided Prompting vs. Chain-of-Thought), demonstrating that each agent remains essential regardless of the prompting method used. Such testing has not been implemented or reported in previous research on counterspeech generation.
>
> **(d) Curated Dataset on Diverse Health Misinformation Topics:**
> A new dataset is introduced, featuring health misinformation posts related to COVID-19, influenza, and HIV, sourced from Reddit and annotated through a combination of classifier-assisted annotation and expert labeling. This dataset is valuable for health misinformation research.
>
>
> ### **W2:** LLMs themselves can be considered as a form of static knowledge base.
>
> **R2:** Thank you for this thoughtful point. We agree that LLMs indeed encode vast amounts of factual information in their parameters. While LLMs can recall general information, prior work shows that they are susceptible to hallucinations [(Huang et al., 2025)](https://doi.org/10.1145/3703155). This limitation motivates the integration of explicit static retrieval in our framework to ensure evidence-grounded responses.
>
> Our experiments also show that the counterspeech generated by prompting without RAG is much inferior in informativeness and factual accuracy, demonstrating the importance of purposeful use of external static knowledge.
>
> **Reference**
>
> Huang, L., Yu, W., Ma, W., Zhong, W., Feng, Z., Wang, H., Chen, Q., et al. (2025). *A survey on hallucination in large language models: Principles, taxonomy, challenges, and open questions*. ACM Transactions on Information Systems, 43(2), 1–55. [https://doi.org/10.1145/3703155](https://doi.org/10.1145/3703155)

---

> > ### Author Response · Authors · 2025-06-02
> > **Additional Updates**
> >
> > ### **W3:** Explanations regarding the design of the proposed framework.
> >
> > **R3:** Thank you for this thoughtful observation. We agree that further elaboration on the design choices within our framework would strengthen the clarity and generalizability of the work. Our current configuration of four agents (static and dynamic retrieval, evidence summarization, generation, and refinement) was motivated by the need to decouple distinct functional tasks and allow targeted optimization and error analysis at each stage ([Chen et al., 2025](https://aclanthology.org/2025.naacl-long.169/);[ Roth et al., 2025](https://arxiv.org/abs/2503.22931)). Modularity is especially valuable in generating counterspeech to health misinformation, as it allows the process to be grounded in accurate evidence, ensures clarity in summarization, and enables the adjustment of tone to suit different contexts.
> >
> > We appreciate your suggestion regarding experimenting with different configurations. While our paper has a four-agent setup, the architecture is inherently flexible and extensible. More work can be done in the future to experiment with fewer or more agents. In our preliminary ablation experiments (Table 2), reducing the number of agents tended to degrade response quality. We will consider more experiments with fewer agents by merging retrieval and summarization, or generation and refinement, or with more agents.
> >
> > We will revise the paper to include the design rationale and discuss the potential for alternative configurations in future work.
> >
> >
> > ### **W4:** The performance gain of the proposed framework is limited compared with the cost.
> >
> > **R4:** Thank you for raising this important point. We acknowledge that our multi-agent framework introduces additional computational and implementation complexity compared to simpler models. However, we believe the performance improvements, particularly in key dimensions like politeness (0.88), informativeness (0.78), and factual accuracy (0.86), demonstrate meaningful gains in quality, especially in the sensitive domain of health misinformation, where trustworthiness and clarity are critical.
> >
> > To clarify, the average processing time per agent for handling one misinformation post is as follows: Dynamic Retrieval Agent: 35.56 seconds, Static Retrieval Agent: 32.43 seconds, Summarization Agent: 38.21 seconds, Counterspeech Agent: 12.80 seconds, and Refinement Agent: 12.05 seconds, leading to a total average time of 131.11 seconds per post. While our multi-agent pipeline introduces additional computational steps compared to simpler models, we argue that the total processing time remains within a reasonable range for public health communications.
> >
> > **Reference**
> >
> > Chen, J., Liang, J., & Wang, B. (2025). *Smurfs: Multi-Agent System using Context-Efficient DFSDT for Tool Planning*. In Proceedings of the 2025 Conference of the North American Chapter of the Association for Computational Linguistics: Human Language Technologies (Volume 1: Long Papers), 1–13.
> >
> > Roth, N., Hidey, C., Spangher, L., Arnold, W. F., Ye, C., Masiewicki, N., Baek, J., Grabowski, P., & Ie, E. (2025). *Factored Agents: Decoupling In-Context Learning and Memorization for Robust Tool Use*. arXiv preprint arXiv:2503.22931. [https://arxiv.org/abs/2503.22931](https://arxiv.org/abs/2503.22931)

---

> > > ### Comment · Reviewer_dAts · 2025-06-09
> > >
> > > Thank you for the reply. I have updated my score regarding your reply.

---

### Official Review · Reviewer_FM46 · 2025-05-13

**Rating:** 6
**Confidence:** 4
**Ethics Flag:** 1

**Summary:**

This paper introduces a multi-agent framework for counter-speech generation, which targets at health misinformation. The motivation is that there exists static knowledge - which are verified and relatively stable, as well as dynamic knowledge that are continuously updated and more up-to-date. By leveraging multiple RAG-based LLMs specialized in different tasks, the authors claim the generation task could be improved on politeness and informativeness. For experiments, the authors collect reddit posts related to health misinformation. The LLMs are prompt-tuned with hand-crafted instructions. Both automatic and human evaluation are done to show the proposed system performs better than single agent baselines, although the differences are rather small.

**Questions To Authors:**

- Have you verified how much of the dynamic knowledge source overlap with the static one?

**Reasons To Accept:**

- The proposed system integrates static and dynamic knowledge in a RAG setup, which seems quite generally applicable to a wide range of applications related to knowledge-based text generation.
- The experiments show improved performance alongside ablation study to further prove the effectiveness of the multi-agent setting.
- Paper is well-written and easy to follow.

**Reasons To Reject:**

- The proposed system requires manual prompt-tuning on each agent. This is likely costly and harder to generalize than a more end-to-end approach.
- The boundary of static and dynamic knowledge might not be as clear cut as it is described in the paper, especially for emerging events. It is not clear how much of the "dynamic" knowledge are citing static knowledge.
- Experiment would be stronger if tested on more than one dataset, especially some of the metrics are quite close in comparison.

---

> ### Author Response · Authors · 2025-06-02
>
> ### **W1:** Manual prompt-tuning on each agent, costly and hard to generalize.
>
> **R1:** We have tested the same model settings across health misinformation posts on COVID-19, influenza and HIV; all consistently show improved performance, which proves the generalizability of the model.
>
>
> ### **W2:** The boundary of static and dynamic knowledge / Overlap of the dynamic knowledge source with the static one.
>
> **R2:** Thank you for the insightful comment. We acknowledge that, particularly during emerging events, dynamic sources may partially cite or echo established (static) knowledge, which can blur the boundary between the two.
>
> In our framework, we define **static knowledge** as rigorously curated, peer-reviewed content from trusted sources such as the CDC, WHO, and NIH. In contrast, **dynamic knowledge** refers to real-time information retrieved from news outlets, health updates, and online fact-checking sources via the web (e.g., using the DuckDuckGo API).
>
> To empirically assess the potential overlap between the two sources, we computed the BLEU score [(Papineni et al., 2002)](https://aclanthology.org/P02-1040/) between retrieved static and dynamic evidence across our samples. The results show a mean **BLEU score of 0.0052 with a standard deviation of 0.0033**, indicating minimal lexical overlap. This suggests that while dynamic knowledge may occasionally cite static sources, it generally provides distinct and complementary information, supporting the separation used in our system design.
>
> We will clarify this distinction in the revised version and explicitly discuss the implications of this overlap.
>
> ### **W3:** Generalization experiments.
>
> **R3:** Thank you for your valuable insight. We conduct generalization experiment using  **MisinfoCorrect**  [(He et al., 2023)](https://dl.acm.org/doi/10.1145/3543507.3583289), which contains misinformation from a different platform Twitter (Now X).  The results in the table below demonstrate that our proposed method generalizes well, achieving higher performance compared to baselines. We will add these results to the revised version.
>
> | **Method**             | **Politeness**  | **Relevance**   | **Informativeness** | **Factual Accuracy** |
> | ---------------------- | --------------- | --------------- | ------------------- | -------------------- |
> | **LLM w DP**           | 0.44 (0.88)     | 0.70 (0.14)     | 0.77 (0.11)         | 0.81 (0.21)          |
> | **LLM w PE**           | 0.81 (0.18)     | 0.77 (0.11)     | 0.77 (0.07)         | 0.94 (0.12)          |
> | **Static RAG**         | 0.86 (0.14)     | 0.77 (0.12)     | 0.78 (0.09)         | 0.95 (0.12)          |
> | **Dynamic RAG**        | 0.85 (0.16)     | 0.77 (0.13)     | 0.76 (0.10)         | 0.92 (0.16)          |
> | **Multi-Agent (Ours)** | **0.89 (0.11)** | **0.78 (0.10)** | **0.83 (0.12)**     | **0.96 (0.12)**      |
>
>
> **Reference:**
>
> Papineni, K., Roukos, S., Ward, T., & Zhu, W. J. (2002, July). *BLEU: a method for automatic evaluation of machine translation*. In *Proceedings of the 40th annual meeting of the Association for Computational Linguistics* (pp. 311-318). [ACL Anthology](https://aclanthology.org/P02-1040/)
>
> He, B., Ahamad, M., & Kumar, S. (2023, April). *Reinforcement learning-based counter-misinformation response generation: A case study of COVID-19 vaccine misinformation*. In *Proceedings of the ACM Web Conference 2023* (pp. 2698–2709). [https://doi.org/10.1145/3543507.3583289](https://doi.org/10.1145/3543507.3583289)

---

### Official Review · Reviewer_p6nq · 2025-05-17

**Rating:** 7
**Confidence:** 4
**Ethics Flag:** 1

**Summary:**

In this paper, the authors propose a multi-agent RAG framework for generating counterspeech against health misinformation. The proposed method integrates LLM-based agents that perform static and dynamic retrieval, summarization, response generation, and refinement to produce counter responses. The static retrieval agent leverages offline knowledge base, while the dynamic retrieval agent performs real-time web search. Another summarization agent condenses the combined evidence to avoid lengthy context, followed by the generation and refinement stages. Experiment results on COVID-19, influenza and HIV misinformation show that the multi-agent system outperforms inference, prompt engineering, static RAG and dynamic RAG baselines. Additional analysis demonstrate the efficacy of the proposed components, and human evaluation confirms that the proposed framework enhances response quality.

**Reasons To Accept:**

(1) Paper is well motivated, easy to follow and well written

(2) The authors proposed a reasonable and effective workflow for multi-agent counter speech generation

(3) The proposed method is evaluated against multiple baselines across different metrics like politeness and factuality, human evaluation is also performed to validate the effectiveness of the proposed method

**Reasons To Reject:**

(1) The paper does not have substantial novelty within the proposed method

(2) Evaluation is not performed for each of the dataset / topics, I wonder how the proposed method perform for different topics

---

> ### Author Response · Authors · 2025-06-02
>
> ### **W1:** Novelty
>
> **R1:** Thank you for your comment. While our approach builds on existing techniques, we believe our contribution lies in the following aspects, which have intellectual merits and practical implications.
>
>
>
> **(a) Integration of Static and Dynamic Knowledge:**
> The framework combines both static (e.g., curated medical guidelines) and dynamic (e.g., real-time web search) evidence to generate counterspeech, enabling responses that are both authoritative and up to date. To our knowledge, this combination has not been explored before, especially in the generation of counterspeech to health misinformation. Our experiments show the significance of including both static and dynamic knowledge.
>
>
>
> **(b) Modular Multi-Agent Pipeline Specialized for Counterspeech Generation Against Health Misinformation:**
> This pipeline presents a modular multi-agent framework tailored for generating effective counterspeech in response to health misinformation. While individual components such as RAG have been studied before, our contribution lies in the integration of dedicated sub-agents responsible for:
> (1) static and dynamic retrieval,
> (2) evidence summarization,
> (3) counterspeech generation, and
> (4) iterative response refinement.
>
> Each agent functions autonomously yet collaboratively within a scalable architecture, enhancing the quality of the generated counterspeech. Experiments show each agent contributes to the improved quality, relevance, and factual grounding in the generated counterspeech. The structured pipeline with these four agents has not been previously applied or evaluated for the generation of counterspeech to health misinformation.
>
>
> **(c) Ablation Studies with Prompting Strategy Comparison:**
> Comprehensive ablation studies evaluate the contribution of each agent in the pipeline. The system is tested under different prompting strategies (e.g., Guided Prompting vs. Chain-of-Thought), demonstrating that each agent remains essential regardless of the prompting method used. Such testing has not been implemented or reported in previous research on counterspeech generation.
>
> **(d) Curated Dataset on Diverse Health Misinformation Topics:**
> A new dataset is introduced, featuring health misinformation posts related to COVID-19, influenza, and HIV, sourced from Reddit and annotated through a combination of classifier-assisted annotation and expert labeling. This dataset is valuable for health misinformation research.

---

> > ### Author Response · Authors · 2025-06-03
> > **Additional Updates**
> >
> > ### **W2:** Evaluation across datasets/topics
> >
> > **R2:** Thank you for the comment. Our dataset covers diverse topics, including COVID-19, HIV, and Influenza. We aggregated the evaluation results across these topics in the following table. The findings show that our proposed method achieves consistently strong average performance across all three topics and all evaluation dimensions, with the following scores: politeness (0.90), relevance (0.71), informativeness (0.77), and factual accuracy (0.85).  We will include this result in the revised version of our paper.
> >
> > | **Method**             | **Category** | **Politeness**  | **Relevance**   | **Informativeness** | **Factual Accuracy** |
> > | ---------------------- | ------------ | --------------- | --------------- | ------------------- | -------------------- |
> > | **LLM w DP**           | COVID-19     | 0.45 (0.27)     | 0.74 (0.17)     | 0.75 (0.14)         | 0.78 (0.21)          |
> > |                        | HIV          | 0.57 (0.31)     | 0.78 (0.12)     | 0.76 (0.06)         | 0.75 (0.27)          |
> > |                        | Influenza    | 0.42 (0.20)     | 0.81 (0.10)     | 0.75 (0.08)         | 0.70 (0.24)          |
> > |                        | **Average**  | 0.48 (0.26)     | 0.78 (0.13)     | 0.75 (0.09)         | 0.74 (0.24)          |
> > | **LLM w PE**           | COVID-19     | 0.78 (0.20)     | 0.69 (0.12)     | 0.76 (0.06)         | 0.81 (0.24)          |
> > |                        | HIV          | 0.83 (0.17)     | 0.71 (0.12)     | 0.75 (0.00)         | 0.79 (0.17)          |
> > |                        | Influenza    | 0.86 (0.13)     | 0.73 (0.08)     | 0.76 (0.06)         | 0.71 (0.25)          |
> > |                        | **Average**  | 0.82 (0.17)     | 0.71 (0.11)     | 0.76 (0.04)         | 0.77 (0.22)          |
> > | **Static RAG**         | COVID-19     | 0.75 (0.18)     | 0.67 (0.12)     | 0.78 (0.08)         | 0.89 (0.17)          |
> > |                        | HIV          | 0.81 (0.13)     | 0.70 (0.17)     | 0.76 (0.06)         | 0.68 (0.35)          |
> > |                        | Influenza    | 0.75 (0.16)     | 0.73 (0.10)     | 0.76 (0.06)         | 0.75 (0.26)          |
> > |                        | **Average**  | 0.77 (0.16)     | 0.70 (0.13)     | 0.77 (0.07)         | 0.77 (0.26)          |
> > | **Dynamic RAG**        | COVID-19     | 0.84 (0.23)     | 0.62 (0.20)     | 0.70 (0.10)         | 0.88 (0.19)          |
> > |                        | HIV          | 0.91 (0.12)     | 0.70 (0.14)     | 0.73 (0.08)         | 0.84 (0.27)          |
> > |                        | Influenza    | 0.82 (0.23)     | 0.69 (0.12)     | 0.74 (0.10)         | 0.66 (0.26)          |
> > |                        | **Average**  | 0.86 (0.19)     | 0.67 (0.15)     | 0.72 (0.09)         | 0.79 (0.24)          |
> > | **Multi-Agent (Ours)** | COVID-19     | 0.92 (0.05)     | 0.68 (0.17)     | 0.78 (0.08)         | 0.84 (0.19)          |
> > |                        | HIV          | 0.86 (0.14)     | 0.73 (0.09)     | 0.78 (0.08)         | 0.91 (0.17)          |
> > |                        | Influenza    | 0.93 (0.14)     | 0.71 (0.10)     | 0.74 (0.06)         | 0.79 (0.15)          |
> > |                        | **Average**  | **0.90 (0.11)** | **0.71 (0.12)** | **0.77 (0.07)**     | **0.85 (0.17)**      |

---

> > > ### Comment · Reviewer_p6nq · 2025-06-09
> > >
> > > Thanks for the response, I will keep my positive scores

---

### Decision · Program_Chairs · 2025-07-08

**Decision:**

Accept

**Comment:**

This is an interesting paper with a focused contribution.